# Mechanisms of Interaction between Enhancers and Promoters in Three *Drosophila* Model Systems

**DOI:** 10.3390/ijms24032855

**Published:** 2023-02-02

**Authors:** Olga Kyrchanova, Vladimir Sokolov, Pavel Georgiev

**Affiliations:** 1Department of the Control of Genetic Processes, Institute of Gene Biology Russian Academy of Sciences, 34/5 Vavilov St., 119334 Moscow, Russia; 2Center for Precision Genome Editing and Genetic Technologies for Biomedicine, Institute of Gene Biology, Russian Academy of Sciences, 34/5 Vavilov St., 119334 Moscow, Russia

**Keywords:** architectural C2H2 proteins, CTCF, Pita, Su(Hw), boundary, insulator, long-distance interactions, eve, Scr, Abd-B, ftz

## Abstract

In higher eukaryotes, the regulation of developmental gene expression is determined by enhancers, which are often located at a large distance from the promoters they regulate. Therefore, the architecture of chromosomes and the mechanisms that determine the functional interaction between enhancers and promoters are of decisive importance in the development of organisms. Mammals and the model animal *Drosophila* have homologous key architectural proteins and similar mechanisms in the organization of chromosome architecture. This review describes the current progress in understanding the mechanisms of the formation and regulation of long-range interactions between enhancers and promoters at three well-studied key regulatory loci in *Drosophila.*

## 1. Introduction

Differential expression of developmental genes in higher eukaryotes has led to a significant complication of the regulatory systems that control gene expression. Several promoters and dozens of enhancers often control expression of a single gene, and enhancers in some cases are hundreds of thousands of base pairs away from their target promoters [1,2]. Our understanding of chromosome architecture and interactions between enhancers and promoters in higher eukaryotes is changing significantly with the development of methods that allow higher-resolution identification of distant contacts in the genome [3,4]. New research methods make it possible to study in more detail the architecture of chromosomes in the nucleus and how long-distance interactions between regulatory elements form. With the appearance of CRISPR/Cas9 technology, approaches to genome editing have been greatly simplified, and every DNA sequence can thus be added or deleted in the regulatory region of interest in vivo [5].

At present, *Drosophila* is the most convenient model object for studying the mechanisms of the formation of chromosome architecture common to all higher eukaryotes. Genome editing techniques can effectively be used in *Drosophila* to easily change particular genes and regulatory sequences [6]. Thus, it is possible to study the functions of every gene and to create complex model systems in vivo. The small size of the *Drosophila* genome facilitates high-resolution genome-wide studies, which yield more accurate results.

The first part of the review gives a brief description of the putative models of long-range interactions between enhancers and promoters. The second part describes the three well-studied *Drosophila* regulatory systems at the *eve* locus and the *Bithorax* and *Antennapedia* gene complexes.

## 2. Models of Distance Interactions between Regulatory Elements

Two models have recently been proposed to explain long-range interactions in the genome. The main model is based on the findings that originate from mammalian Hi-C and ChIP-seq studies and indicate that the cohesin complex, together with CTCF, forms most of the enhancer–promoter interactions and boundaries of topology-associated domains (TADs) [7,8,9,10]. Inactivation of the cohesin complex or CTCF results in partial disruption of chromosome organization in TADs [11,12,13]. The cohesin complex is highly conserved in eukaryotes, and its main function is to hold sister chromatids together during mitosis and meiosis [14,15]. The cohesin complex consists of four subunits, which form a ring around the two DNA strands by using the energy of ATP [15]. A cluster consisting of 11 zinc finger domains of the C2H2 type is a feature of the structure of the CTCF protein [16,17,18]. Five C2H2 domains of CTCF specifically bind to a 15 bp motif, which is conserved in animals and determines most of the functional properties of this architectural protein [19]. A conserved motif interacting with the cohesin complex was found at the N-terminus of human CTCF [20]. A classical model suggests that, once fixed on chromatin, the cohesin complex begins ATP-dependent DNA extrusion with the formation of a chromatin loop [21]. CTCF blocks the movement of the cohesin complex, thus leading to fixation of the boundaries of chromatin loops at the CTCF sites [22].

An alternative group of models is based on the studies of mammalian LIM domain-binding factor 1 (LDB1) [23], the *Drosophila* architectural C2H2 proteins [24,25], and the *Drosophila* proteins that preferentially regulate the activity of housekeeping promoters [26,27].

In mammals, the C-terminal domain of LDB1 interacts with DNA-binding transcription factors of the LIM family [23]. The N-terminal domain of LDB1 forms a stable homodimer [28] to maintain long-range interactions between enhancers and gene promoters [29,30].

In *Drosophila*, several architectural C2H2 proteins have been characterized and shown to preferentially bind to gene promoters and known insulators [17,24,25]. The architectural proteins of this group have clusters of C2H2 domains, some of which specifically bind to motifs of 12 to 18 bp in length [17,24,25]. Most of the *Drosophila* C2H2 architectural proteins have structured domains that form homodimers at the N-terminus [31,32,33]. Interestingly, unstructured homodimerization domains are found at the N-terminus in the CTCF proteins of various animals, including *Drosophila* and mammals [34]. The domain is required for functional activity of *Drosophila* CTCF (dCTCF) [35], while the role of similar domains in mammalian CTCFs remains unstudied. In *Drosophila*, dCTCF, Pita, and Su(Hw) are the best-characterized architectural C2H2 proteins and determine the activity of most of the known *Drosophila* insulators [36,37,38]. Binding sites for these proteins can support long-distance interactions between regulatory elements in model transgenic lines [33,39,40].

The CP190, Chromator, Z4, and BEAF proteins preferentially bind to insulators and promoters of housekeeping genes, which are at the boundaries of most *Drosophila* TADs [26,27,41,42,43]. The proteins interact with each other and contain homodimerization domains [44,45,46,47,48], suggesting their likely involvement in maintaining long-distance interactions. Like mammalian LDB1, CP190 is recruited to regulatory elements through interactions with DNA-binding transcription factors including dCTCF, Pita, and Su(Hw) [49].

Either model by itself cannot explain a number of experimental results. For example, it was shown using Micro-C that inactivation of CTCF or cohesin does not affect the formation of chromatin loops between regulatory elements in mouse embryonic stem cells [50]. On the other hand, alternative models do not explain how distant chromatin regions initially find each other to form a stable pairing, which is necessary for the organization of chromatin loops. The most obvious is a combination of the two models, which will explain most of the current experimental data in both mammals and *Drosophila* (Figure 1A). In *Drosophila*, ChIP-seq data show that motifs recognized by different architectural C2H2 proteins are combined in many insulators and promoters [51,52]. Recent studies in mammals showed that, in well-studied genomic regions, CTCF binds in cooperation with the other C2H2 proteins ZNF143, MAZ, and WIZ [53,54,55,56], which are involved in the formation of long-distance interactions. MAZ and WIZ were shown to interact with the cohesin complex [54,55]. The cohesin complex likely interacts with a large number of C2H2 proteins. It can be assumed that the movement of cohesin complexes is most efficiently blocked in the chromatin regions that are associated with groups of C2H2 proteins. As a result, cohesin brings the regulatory elements together in a space, and their pairing is additionally stabilized by multiple interactions between the homodimerized domains of C2H2 architectural proteins and their associated partner proteins, such as CP190, Z4, and Chromator.

The specificity and stability of the interaction between two regulatory elements is determined by the number of involved proteins whose domains are capable of forming homodimers (Figure 1A). Studies in transgenic *Drosophila* lines showed that two identical copies of any of the insulators tested pair in a head-to-head orientation [39,57]. When two identical insulators were oriented head-to-head, the configuration of the resulting chromatin loop was favorable for the interaction between a promoter and an enhancer located outside the loop (Figure 1B). When the insulators were in the same orientation, the enhancer could only stimulate the promoter when it was inside the loop. Such orientation-dependent interaction between identical copies of insulators is consistent with the model that regulatory elements consist of binding sites for several C2H2 architectural proteins, each of which can support long-distance interactions via its homodimerization domains. A direct consequence of the model is that inactivation of any architectural protein should not significantly affect the organization of chromosome architecture but may disrupt the individual local interactions between enhancers and promoters.

## 3. Current Models of Enhancer—Promoter Communication

Enhancers usually average about 500 bp in size and consist of combinations of motifs recognized by DNA-binding transcription factors (TFs), which suppress or activate enhancer activity (Figure 2A). [58]. Enhancers can be assembled into large modular super enhancers, which range in size from 5 to 50 kb [59]. The main function of enhancers is to mediate the recruitment of the mediator complex to promoters, resulting in transcriptional activation [60,61].

The mediator complex is conserved in eukaryotes and consists of 26 subunits in mammals. The subunits are grouped in three modules, which are called the head, middle and tail (Figure 2A). A core part of the mediator interacts with the kinase module, which can function both as part of the complex and separately [60]. The head and middle modules provide interaction with RNA polymerase II; the tail module is responsible for the binding of the mediator with TFs on enhancers and the main TFIID complex on promoters [61] (Figure 2B). Binding to the mediator complex, the kinase module blocks its interaction with RNA polymerase II. The tail module is the most flexible and can take on various conformations [62,63]. The mediator complex binds to the non-phosphorylated carboxy-terminal domain (CTD) of RNA polymerase II, and the binding changes the conformation of the tail module. Next, RNA polymerase II is released from the complex with the mediator after CTD phosphorylation on the promoter to change the conformation of the tail module again. It is likely that different conformations of the tail module determine the specificity of binding of the mediator with TFs on enhancers or TFIID on promoters.

Several complexes with enzymatic activities are also recruited to enhancers: acetyltransferase (p300/CBP), methyltransferase (Mll3/Mll4/COMPASS), and deubiquitinase [64]. Mll3/Mll and p300/CBP are responsible for histone H3 monomethylation at lysine 4 (H3K4me1) and acetylation at lysine 27 (H3K27ac), respectively. The H3K27ac and H3K4me1 modifications of histone H3 are thought to reduce the stability of nucleosomes, resulting in the formation of open chromatin [65]. In addition, the enzymatic complexes can introduce modifications into TFs that bind to enhancers and gene promoters, thereby stimulating their activity [66,67]. For example, p300/CBP may play an important role in acetylation of the TFs involved in the pre-initiation complex formation [66]. Acetylation of different domains in the p53 protein usually positively regulates its activity [68]. Methylation of p53 at K327 increases its stability and ability to stimulate transcription [69]. There are other examples of the positive role of TF methylation and acetylation, but this area remains poorly studied in general.

In addition to transcription activators, repressors are recruited to enhancers to suppress their activity in cells where the enhancers should not function (Figure 2A). Complexes with deacetylase and, less commonly, demethylase activities are recruited to enhancers by repressors [70]. Deacetylation of TFs on enhancers probably decreases their ability to attract enzymatic and mediator complexes. In addition, histone deacetylation increases chromatin compaction, thereby reducing the ability of TFs to bind enhancers [71]. Thus, enhancer activity in a particular cell is determined by the concentration of TFs interacting with activator and repressor complexes (Figure 2A).

The Polycomb proteins play an important role in the suppression of enhancer activity [71,72,73]. Two main Polycomb complexes are known in *Drosophila*, of which one has ubiquitinating activity (Polycomb repression complex 1, PRC1) and the other has methyltransferase activity (Polycomb repression complex 2, PRC2) [73]. PRC1 and PRC2 can be recruited directly to enhancers and promoters through interactions with DNA-binding TFs [74]. A large number of variations in these two basic Polycomb complexes have been found in mammals, with them being determined by the need to finely regulate numerous groups of enhancers and promoters during development and cell differentiation [71]. The most studied mechanism of repression is the formation of inactive chromatin through the introduction of H3K27me3 and H2AK119ub modifications into nucleosomes mediated by Polycomb complexes [71,73]. Methylation and ubiquitination of key TFs is also a possible mechanism to suppress enhancers and promoters. For example, methylation of lysine 99 in the coactivator BRD4 negatively regulates its activity in transcription [75].

Recruitment of the Polycomb complexes to enhancers can lead to their transformation into silencers that repress transcription of adjacent genes [76,77,78]. *Drosophila* has well-characterized, specialized regulatory elements that specifically recruit PRC1 and PRC2 and they are called Polycomb response elements (PREs) [79]. Such regulatory elements can function as specific silencers, increasing the efficiency of the complete repression of the enhancers and promoters that should be completely turned off in a certain group of cells during development [80,81].

Two recent studies [82,83] investigated the compatibility of enhancers and promoters. It was found that enhancers preferentially activate weak promoters rather than strong promoters, which normally determine the transcription of housekeeping and cell cycle genes. In general, it was shown that most of the enhancers tested can activate almost every promoter. A lack of specificity of interactions between enhancers and promoters presumably increases the role of insulators and TADs in limiting enhancer–promoter interactions.

However, recent studies have shown that TADs do not block long-range interactions between enhancers and promoters [50]. It was shown using *Drosophila* transgenic model systems that chromatin loops formed by interacting insulators cannot effectively block the interaction between enhancers and promoters [40,84,85]. Thus, there are no strict structural restrictions to block the co-localization of enhancers and promoters belonging to different regulatory domains. Using micro-C, intense contacts were detected in the genome between certain genomic sites including enhancers, promoters, and insulators that do not coincide with TAD boundaries [50,86,87,88]. A special class of regulatory elements, called tethering elements, was isolated in *Drosophila*. The elements occur next to enhancers and promoters and form stable chromatin loops between them [86]. Ultra-high resolution microscopy showed that some functionally interacting enhancers and promoters are relatively far away from each other [3,89].

It can be assumed that mediator complexes are concentrated on enhancers as a result of multiple interactions between subunits of the tail module and unstructured domains of enhancer-associated TFs [61] (Figure 2B). In the next stage, the mediator leaves the enhancer as a result of a change in the conformation of the tail module. Conformational changes in the tail module are possibly a result of methylation (Mll3/Mll4/COMPASS?), acetylation (p300/CBP?), or phosphorylation (the kinase module?) of subunits of the mediator complex. However, this issue has not been studied as of yet. In the new conformation, the tail module has greater affinity for the TFIID complex on the promoter, resulting in pre-initiation complex formation and the recruitment of RNA polymerase II. The enhancer-bound p300/CBP complex can simultaneously acetylate TFs to activate them. Increasing concentrations of active forms of the mediator complex and TFs should stimulate the promoters located in a certain active zone around the enhancer [89,90]. It does not matter to such a trans-activation mechanism whether the enhancer and promoter are in close contact, interact briefly, or are at some distance from each other. Interactions between insulators and/or tethering elements lead to the formation of chromatin loops, which form a region in which enhancers stimulate a specific group of promoters. In some cases, chromatin loops can reduce the likelihood of promoter localization in the nuclear region where the enhancer functions.

## 4. Interacting Insulators form an Autonomous Regulatory Domain of the *eve* Gene

The regulation of the pair-rule gene *even-skipped* (*eve*) is one of the best studied in *Drosophila* (Figure 3A). [91,92,93,94]. Eve belongs to a group of primary pair-rule factors whose stripe-pattern expression starts in early embryonic development [95,96]. The *eve* gene is in the center of a 16 kb domain surrounded by housekeeping genes, which are active in all cells.

The body is divided into segments with certain morphological differences in *Drosophila*, like in all insects [97]. Segments formed at the embryonic stage are called parasegments (PSs). During the early development of an embryo, 14 PSs are formed, corresponding to anatomical structures of the larva. PSs are initially determined by the products of the maternal genes *Bicoid* (*Bcd*), *Hunchback* (*Hb*), and *Caudal* (*Cad*), which precisely regulate the expression levels of gap group genes, including *hunchback* (*hb*), *Kruppel* (*Kr*), *knisps* (*kni*), and *giant* (*gt)* [98,99,100,101]. In early embryos, the maternal and gap genes cooperatively regulate the expression of a large group of pair-rule genes, including *eve* and *fushi tarazu* (*ftz*) [102,103]. The *eve* gene is expressed in seven broad stripes along the anteroposterior (AP) axis of the embryo during its early development (Figure 3A). At this stage, *eve* expression is controlled by five enhancers that are active in separate stripes [95,96]. The stripes that express *eve* subsequently become thinner with clear anterior and posterior borders [104]. Expression of the *eve* gene at this stage is controlled by a single enhancer, which is bound with the early pair-rule proteins paired, runt, and sloppy-paired [105]. At late stages of embryonic development, *eve* expression loses its characteristic pattern and is controlled by several tissue-specific enhancers.

The *eve* enhancers contain binding sites for ubiquitous transcriptional activators, such as STAT and Zelda (Zld), and the maternal Bicoid activator [70,106,107,108]. Repression of the enhancers is controlled by the Kr, Kni, and Gt proteins, which recruit the CtBP repressor complex [70]. CtBP-dependent repressor complexes have deacetylase activity. Finally, the Hb protein can recruit activators or repressors to the enhancers, depending on the nearby partner proteins [109]. For example, the stripe 3 + 7 enhancer is stimulated by the activators Zld and STAT and repressed by Hb and Kni [107,108]. At the same time, the stripe 2 enhancer is controlled positively by Zld, Hb, and Bcd and negatively by Gt and Kr.

Each stripe enhancer has a specific set of activator and repressor motifs, which are arranged in a specific sequence and orientation. Each stripe enhancer shows more efficient recruitment of activator (acetylase activity) or suppressor (deacetylase activity) complexes, depending on the concentration of gap repressors in the nucleus. TF acetylation/deacetylation is likely to stabilize the active/inactive status of each stripe enhancer. Deacetylation of nucleosomes also leads to the formation of more stable local chromatin, which blocks the binding of activators to enhancers. This possibility is consistent with the finding that the Zelda and Hb proteins cannot stably bind to their sites on chromatin [110].

The complex regulatory region of the *eve* gene (Figure 3A) is flanked by housekeeping genes, which are expressed in all cells [111,112]. The housekeeping gene *TER94* is on one side of the regulatory region of the *eve* gene and is actively transcribed in all cells. The other side is flanked by the 3′ region of the *CG12134* gene, which shows ubiquitous but weaker expression.

A 368 bp insulator (Figure 3A) was found immediately upstream of the core promoter of the *TER94* gene [111,112]. The insulator efficiently blocks the activity of embryonic enhancers in model transgenic lines. When the insulator was inserted into the P-transposon, the construct was found to preferentially integrate into the genomic region near the *eve* locus [111,112]. This effect is called homing and is explained as follows. When DNA of the P-transposon with the insulator is injected, proteins are assembled on the insulator to form a complex, which interacts with a similar complex on the endogenous insulator to increase the specific integration of the P-transposon. The insulator was therefore named Homie. The function of Homie in vivo is currently unknown since its deletion has not been obtained. It is likely that Homie performs many functions, one of which is to be the distal part of the *TER94* gene promoter since deletion of the insulator significantly reduced *TER94* expression in transgenic lines [111].

A PRE was found next to the insulator (Figure 3A); its function is to negatively regulate the *eve* gene enhancers at the late stages of embryogenesis [111]. Homie was assumed to protect *TER94* expression from the PRE, which represses *TER94* transcription in oocytes and late embryos in transgenic lines [111]. A second insulator (Figure 3A), named new Homie (NHomie), was found between the 3′ UTR of the *CG12134* gene and the regulatory region of the *eve* locus [113]. Interestingly, both insulators are bound with the Su(Hw) [114] and Ibf1/2 [115] proteins. The proteins can be involved in recruiting CP190 and Mod(mdg4)-67.2 to Homie and NHomie [115,116]. Homie additionally binds with Pita, which is another architectural C2H2 protein, and also interacts with CP190 [52,117]. Thus, the Homie insulator has binding sites for two architectural C2H2 proteins. In Micro-C studies, Homie and NHomie efficiently interacted to form a small TAD in embryos [86].

To study the role of the insulators flanking the *eve* locus, a construct was designed to include the entire *eve* locus with neighboring insulators. The *eve* gene was replaced by the *lacZ* reporter and the *TER94* gene was replaced by the *EGFP* reporter. The transgene was integrated into various regions of the genome by using a P-transposon [111] or φC31 integrase system [118]. In most transgene integration sites, the *lacZ* reporter retained a regular strip transcription pattern similar to that of the endogenous *eve* locus. Deletion of either of the two insulators only slightly affected the *lacZ* expression pattern. However, a simultaneous deletion of both insulators significantly affected the formation of an *eve*-like pattern of reporter gene expression. These results suggest that the interaction between the Homie and NHomie insulators modestly increases the efficiency of the interaction between enhancers and the *eve* gene promoter. Interestingly, deletion of the Homie insulator induces expression of the *TER94* gene with *eve*-like patterns [111]. A similar result was observed for the P-element promoter present in the P-transposon and reporter expression driven by the minimal *hsp70* promoter [118]. Thus, the *eve* enhancers can nonspecifically activate various promoters in early embryos. The findings are consistent with the model that an active enhancer induces the spreading of an active Mediator complex and/or acetylated TFs, which stimulate nearby promoters. A possible alternative model is that transient contacts between an active enhancer and neighboring promoters activate the promoters.

The most interesting is the study of the interaction between Homie insulators located in the endogenous locus and a transgene inserted at a distance of 142 kb [113,119]. The interaction between the identical insulators physically brings the transgene and endogenous locus closer together, thus allowing the *eve* enhancers to stimulate the reporter under the control of the minimal *hsp70* promoter (Figure 3B). The paring occurs in a head-to-head orientation [113], which can be explained by homo-interactions between architectural C2H2 proteins bound to both insulators. The mutual orientation of the insulators located in the construct and in the endogenous site determines which of the two reporter promoters is activated by the *eve* enhancers. This finding clearly demonstrates how the interaction between two insulators/tethering elements can facilitate or isolate long-distance enhancer–promoter interactions.

In the endogenous *eve* locus, the interaction between Homie and NHomie brings two housekeeping genes in closer proximity and improves the protection of the TERT promoter from PcG-mediated silencing mediated by a nearby PRE. It is most likely that a chromatin loop formed by the insulators facilitates a functional link between the selected active enhancer and the *eve* promoter. At the same time, the chromatin loop prevents the strong promoters of housekeeping genes from entering the zone of action of the *eve* enhancers.

## 5. Insulators and Tethering Elements Provide Independent Regulation of Genes in the Antennapedia Gene Complex

The Antennapedia gene complex (ANT-C) is one of the two major Hox gene clusters in the *Drosophila* genome. ANT-C controls the development of PS1–PS4, which form the structures of the head, the first thoracic segment (T1), and the anterior compartment of the second thoracic segment (T2) [120]. ANT-C is about 500 kb long and contains five homeotic selector genes: *labial* (*lab*), *proboscipedia* (*pb*), *Deformed* (*Dfd*), *Sex combs reduced* (*Scr*) gene, and *Antennapedia* (*Antp*) [121] (Figure 4A).

An interesting feature of ANT-C is the presence of the pair-rule gene *fushi tarazu* (*ftz*) between the *Scr* gene and its early enhancer (EE), which are separated by 25 kb [122,123,124]. The *ftz* gene is an early pair-rule gene that determines the development of even parasegments in *Drosophila* and shows an expression pattern that is similar to that of the *eve* gene and has a form of seven stripes along the AP axis of the embryo [102]. Transcription of the *Scr* gene begins later during embryogenesis and peaks at the late larval and early pupal stages [120]. At the early embryonic stage, the *ftz* gene is regulated by three enhancers, each of which determines gene expression in two stripes [125,126,127]. In addition, one enhancer combines the activation of the *ftz* gene in the fourth stripe in early embryos and gene activation in all stripes (zebra-like function) during later embryogenesis [128,129]. The mechanisms of *eve* and *ftz* expression are similar in early embryos, and the only obvious difference is that *ftz* is within the regulatory region of *Scr*, which is inactive in early embryogenesis. Two insulators, SF1 and SF2, were found at the boundaries of the *ftz* regulatory region [130]. The insulators were shown to efficiently block the activity of embryonic enhancers in transgenic *Drosophila* lines [131]. The SF1 and SF2 insulators have binding sites for the architectural proteins dCTCF and Pita, respectively [33]. Interestingly, peaks of dCTCF and Pita are found on both insulators in ChIP-seq analysis of embryos, which is likely due to paring between these insulators [132]. The CP190 protein was found on the SF1 and SF2 insulators [133]. CP190 is most likely recruited by dCTCF, Pita, and other as yet unidentified C2H2 architectural proteins that bind to both insulators. The regulatory region of *ftz*, which is active in early embryos, is protected by insulators from the surrounding repressed chromatin enriched in H3K27Me3 and H3K9Me3 histone modifications [130]. The interaction between SF1 and SF2 weakens in 12–16 h embryos, allowing repressive marks to spread in the regulatory region of *ftz* and increasing the influence of surrounding regulatory elements on *ftz* transcription [130]. At the same time, SF1 continues to interact with other insulators identified across the ANT-C regulatory region [130,134,135]. Thus, long-distance interactions between the SF1 and SF2 insulators are regulated during development [136]. Deletion of SF1 or SF2 affects *frz* expression, which becomes partly controlled by the regulatory region of the *Scr* gene [86].

Micro-C analysis has shown that the *Scr* regulatory region is within a 90 kb TAD [86]. The interaction between the distal EE and the *Scr* promoter is supported by the distal tethering element (Scr_DTE), which is 6 kb away from the enhancer [86,127] (Figure 4). Scr_DTE interacts with a 450 bp region (Scr_TE), which is 100 bp away from the transcription start site of the *Scr* promoter [137]. Interestingly, the 450 bp proximal part of the promoter and Scr_DTE contain, respectively, eight and four copies of the TTCGAA palindrome, which is necessary but not sufficient for the functional activity of these regulatory elements [127,137]. The protein that binds to the repeats remains unknown, but both TEs recruit the key early developmental factors Zelda, Clamp, and GAF [86,138,139,140,141,142]. Deletion of Scr_DTE significantly reduces the interaction between the promoter and EE, and this is accompanied by later activation of the *Scr* gene [86]. When Scr_DTE is deleted, communication between the *Scr* promoter and EE is possibly partly maintained by the interaction between the SF1 and SF2 insulators. Deletion of EE does not disrupt the interaction between the TEs. Thus, the TEs form a stable loop that is not regulated by the activity of the nearby EE (Figure 4B).

Interestingly, a similar result was observed in the case of the interaction between the P1 promoter of the *Antp* gene and its EE, which is 38 kb upstream of the gene [86] (Figure 4A). Micro-C analysis showed that TEs that determine the stable interaction between the regulatory elements are directly adjacent to the P1 promoter (P1_TE) and the enhancer (Antp_DTE). These TEs also have binding sites for the proteins Zelda, Clamp, and GAF [138,139,140,141,142]. Deletion of Antp_DTE led to loss of the specific interaction between the promoter and enhancer, thus significantly delaying the activation of gene transcription. However, the level of *Antp* transcription restored over time as in the case of the *Scr* gene. Thus, the interaction between the TEs of the *Antp* and *Scr* loci is not critical in the communication between enhancers and promoters, since the chromatin architecture is simultaneously maintained by interacting insulators, which are usually located at the boundaries of each regulatory domain in ANT-C (Figure 4B).

Deletion of the SF1 or SF2 insulator significantly decreased *Scr* expression but did not disrupt the interaction between the TEs, pointing to the autonomy of the interaction between these elements [86]. Thus, the interactions between the SF1 and SF2 insulators and between the TEs occur independently of each other despite the fact that the chromatin loop formed by the insulators is inside the TE-dependent loop (Figure 4B,C). As with the *eve* locus, the *ftz* regulatory region is shaped by interacting insulators that allow the stripe enhancers to function autonomously from the surrounding repressed chromatin in early embryos. The interacting TEs of the *Scr* and *Ant* genes form stable chromatin loops, and the loop organization is independent of the active/repressed state of neighboring enhancers. Previous studies have shown that many long-distance interactions between enhancers and promoters form before transcription activation and remain stable throughout *Drosophila* embryogenesis [143].

## 6. Boundaries Organize the Enhancer—Promoter Interactions in the *Abd-B* Gene of the Bithorax Complex

The Bithorax complex, BX-C, occupies more than a 300 kb region and consists of nine cis-regulatory domains. The positions of the domains along a chromosome are the same as the positions of the segments that they control: the third thoracic (PS5 (or segment T3 in the adult)) and all abdominal segments of *Drosophila* (PS6–PS13 (A1–A9)) [144,145]. Each domain contains enhancers, which determine the expression pattern of one of the three homeotic genes *Ultrabithorax* (*Ubx*), *abdominal-A* (*abd-A*), and *Abdominal-B* (*Abd-B*) [146,147,148] (Figure 5A). Regulatory domains are flanked by boundaries [149,150,151,152,153,154], which block cross-talk between adjacent domains. Some of them (*Fub*, *Mcp, Fab-6*, *Fab-7*, and *Fab-8*) have been tested in transgene model systems and were shown to have insulator activities [149,152,155,156,157,158]. Deletion of the boundary leads to fusion of the domains and transforms the anterior segment into a copy of the posterior one. All BX-C regulatory domains are organized in a similar way [144,159]. Only a part of the *Abd-B* regulatory region is described in detail here, and this part is currently the best studied.

The expression of *Abd-B* in the A5, A6, and A7 segments is determined by the *iab-5*, *iab-6,* and *iab-7* regulatory domains, respectively [160] (Figure 5A). The functional autonomy of the *Abd-B* regulatory domains is determined by the *Mcp, Fab-6*, *Fab-7,* and *Fab-8* boundaries [144,159]. Each regulatory domain contains a PS-specific element called an initiator, whose activity is under the control of early developmental activators and repressors [161,162,163]. Deletion of the initiator inactivates the regulatory domain [161,164].

In the best studied *iab-5* domain, the initiator is organized by two Ftz and two Kr binding sites, which are closely spaced [165]. Ftz has been shown to act as an activator for *iab-5*, while Kr is responsible for repressing *iab-5* activity in the anterior of the embryo. When one of the Kr sites is mutated, premature activation of the initiator occurs in PS8 of the embryo and is accompanied by a partial transformation of A3 into A5 [166]. At the late stages of *Drosophila* development, *Abd-B* expression is regulated, in particular, by two partially overlapping tissue-specific enhancers [167], which determine the pigmentation of the A5 segment and a reduced density of trichomes on the tergite in males.

The *Mcp* boundary (Figure 5A) separates the *abd-A* and *Abd-B* regulatory regions and determines the autonomy of the *iab-5* domain [151]. *Mcp* deletion allows the *iab-4* initiator to induce premature activation of the *iab-5* domain, thus leading to stimulation of *Abd-B* in the A4 segment and, consequently, transformation of the A4 segment into A5. The *Mcp* insulator was mapped to a 430 bp region, which contains binding sites for the architectural proteins Pita and dCTCF [158,168] (Figure 5A). The 210 bp *Mcp* core including the dCTCF and Pita motifs only partially retains insulator activity, but can support long-range interactions between transgenes [51,169,170]. A PRE is next to *Mcp* and negatively regulates the *iab-5* enhancers, restricting *Abd-B* activation [171].

The *Fab-6* boundary separates the *iab-5* and *iab-6* domains and consists of two nuclease-hypersensitive regions HS1 + HS2 [153,164,172] (Figure 5A). The central part of the boundary, including two dCTCF binding sites, functions only as a weak insulator [153,172]. Surprisingly, in vivo deletion analysis showed that the functional boundary consists of the insulator (HS1) and the nearby PRE (HS2) [172]. It is of interest that the core part of the *Fab-6* insulator displays the properties of a Polycomb-dependent repressor in transgenic lines [153]. Thus, the PRE and insulators can cooperate in the formation of independent regulatory domains of *Abd-B*.

The complete *Fab-7* boundary separates the *iab-6* and *iab-7* domains and consists of four nuclease-hypersensitive regions HS* + HS1 + HS2 + HS3 [151,173,174] (Figure 5A). The HS2 region contains two Pita binding sites [168], and seven GAF binding sites are found in the HS1 region [173]. HS3 is a PRE that acts as a suppressor of tissue-specific enhancers in the *iab-7* domain [174]. Paring between identical copies of *Fab-7* can support long-range interactions between two transgenes [156,170,175] or between a transgene and BX-C [176]. In vivo deletion analysis showed that the insulator function can be reproduced by the distal part of HS1 (dHS1) and HS3 (PRE), which are individually weak insulators [173,177]. The HS*, HS1, and HS2 regions individually also have only weak insulator activity, which is not fully restored even when they are placed together [177]. Thus, as with the *Fab-6* boundary, the PRE plays a role in organization of the functional *Fab-7* boundary.

The *Fab-8* boundary separates the *iab-7* and *iab-8* domains [149,157]. The functional insulator was localized to a 337 bp region, which contains two CTCF sites [178] (Figure 5A). Thus, like the *Mcp* boundary, *Fab-8* is compact and does not require a PRE for the insulator function.

Mapping of the functional regions in *Fab-8* and *Mcp* showed that at least two additional unknown architectural proteins must bind to 337 bp *Fab-8,* which contains two dCTCF sites, and 340 bp *Mcp,* which contains Pita and dCTCF sites, to form an insulator [51]. Artificial sites consisting of 4–5 motifs for one of the three architectural proteins dCTCF, Su(Hw), and Pita can also function as efficient insulators between the *iab-6* and *iab-7* domains [51,178,179]. These architectural proteins are able to recruit CP190 to the boundaries. Expression of a mutant Pita protein unable to interact with CP190 abolished insulator activity at the Pita binding sites [49]. CP190 interacts with Z4 and Chromator, which can function together in blocking crosstalk between the *iab* regulatory domains. It can be assumed that the efficiency of insulation directly depends on the number of CP190 complexes recruited to the boundary.

Not only do the *Fab-6*, *Fab-7,* and *Fab-8* boundaries block crosstalk between the *iab* domains, but they also support specific long-distance interactions between the enhancers and *Abd-B* promoters [178,179,180,181]. In contrast, artificial insulators consisting of 4–5 motifs for architectural proteins function only as insulators [51,168]. An addition of about 150 bp regions from the *Fab-7* or *Fab-8* boundaries to the artificial insulators was found to restore proper activation of *Abd-B* by the *iab* enhancers [179,180]. Moreover, a substitution of *Mcp* with such chimeric boundaries facilitates ectopic activation of *Abd-B* by the *iab-4* enhancers in the A4 segment [181]. It was speculated that the approximately 150 bp regions function as tethering elements by interacting with similar regions in the *Abd-B* promoter region. This model is indirectly supported by the interaction observed for the *Fab-7* or *Fab-8* boundaries with the *Abd-B* promoter in micro-C studies in embryos [86]. The *Fab-7* and *Fab-8* tethering elements bind with the late boundary complex (LBC) [173,179,180,182]. An interesting feature of this complex is the ability to specifically bind to long sequences of 50–60 bp that contain several short characteristic motifs. All three currently known subunits of the complex—CLAMP, Mod(mdg4), and GAF [179,180,183]—have N-terminal homodimerization domains. Mod(mdg4) and GAF contain BTB domains, which form homohexamers [44,184]. Like CTCF, CLAMP has an unstructured domain that can be homodimerized [185]. It is possible to assume that LBCs can support specific long-range interactions due to a large number of homodimerization domains that form the complex. Interestingly, several regions have been identified in the aria of the *Abd-B* promoters, to which GAF, Mod(mdg4), and CLAMP bind simultaneously [138,139,140,141,142]. One of these regions interacts with the boundaries, as evidenced by micro-C analysis of embryos [86].

According to the most probable model, stimulation of *Abd-B* expression in the corresponding *iab* domain is initially determined by activation of the initiator located in this domain (Figure 5B). Next, the initiator stimulates the corresponding boundary to form a contact with the *Abd-B* promoter region. A chromatin loop formed between the boundary and the promoter region allows the *iab* enhancers to activate *Abd-B* transcription.

## 7. Conclusions

In all three *Drosophila* loci described here, insulators or tethering elements are responsible for organizing the long-distance interactions that bring functionally interacting enhancers and promoters closer together in space. The distinction between insulators and tethers remains unclear. Like tethering elements, insulators can be an integral part of a promoter in some cases. The only difference is that insulators function to block the local interactions between regulatory elements and thus form a boundary between chromatin regions enriched in nucleosomes with active and repressive histone modifications. Stable long-range interactions that exist between regulatory elements in most cells for a long time are efficiently detected in genome-wide studies. However, examples of enhancer–promoter communication in BX-C and ANT-C show that some of the long-range interactions form only upon activation of an enhancer or at a certain stage of *Drosophila* development. It is likely that most of the regulated long-distance interactions remain undetected in genome-wide studies of whole organisms. The mechanisms that regulate the long-distance interactions are currently poorly understood [38]. Of interest is the discovery of the RNA-binding protein Sherp, which suppresses the interaction between enhancers and promoters [62] and the activity of insulators [186,187] in the nervous system.

It is now becoming clear that similar mechanisms underlie the long-distance interactions in mammals and *Drosophila* [188]. In both mammals and *Drosophila*, CTCF needs partner proteins to form chromatin loops along with cohesin, and most local interactions are independent of CTCF and cohesion [50,53,54,55,56]. Thus, *Drosophila* provides a convenient model to study the general principles and mechanisms that determine the formation and regulation of long-distance interactions in animals.

Several studies have shown that problems in the formation of chromosome architecture play a significant role in disrupting the regulatory programs of cells and may cause human diseases [189,190,191]. To develop therapeutic agents that prevent the consequences of chromosome architecture disorders, it is necessary to study in detail the properties and mechanisms of functioning of all architectural proteins. The possibility to efficiently make necessary changes to the *Drosophila* genome creates conditions for a faster and more efficient study of the properties of architectural proteins than is currently possible in mammals. The relatively small *Drosophila* genome makes it possible to identify and to study all of the main architectural proteins in the near future. This is necessary for understanding the mechanisms that form the architecture of animal chromosomes.

## Figures and Tables

**Figure 1 ijms-24-02855-f001:**
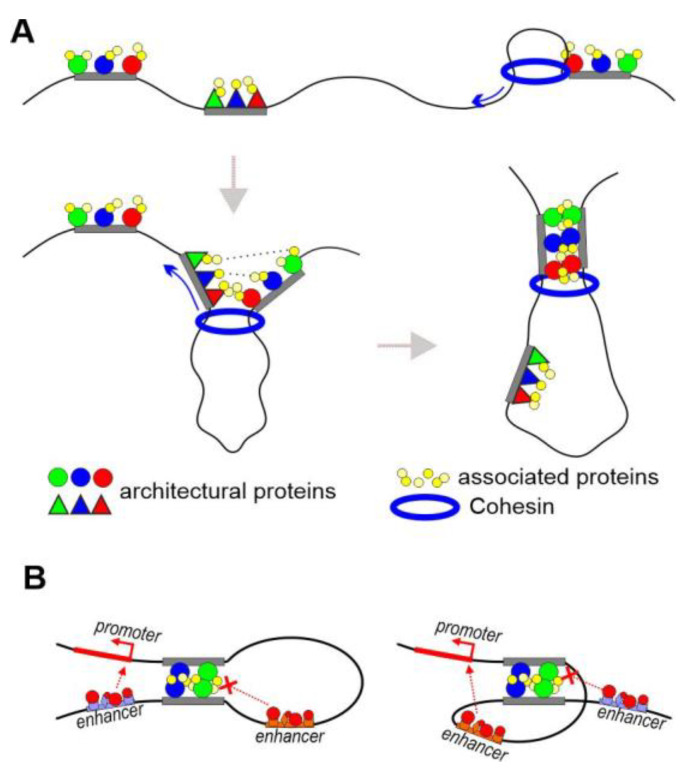
Combination of two models of distance interactions. (**A**) Local interaction between regulatory elements. Various combinations of architectural proteins bind to insulators or tethering elements. The same associated proteins (such as CP190, Z4, and Chromator) bind to different combinations of architectural proteins. The specificity of distance interactions between tethering elements/insulators is determined by the number of C2H2 proteins associated with different elements that are capable of interacting with each other. (**B**) Two copies of an insulator interact in head-to-head orientation.

**Figure 2 ijms-24-02855-f002:**
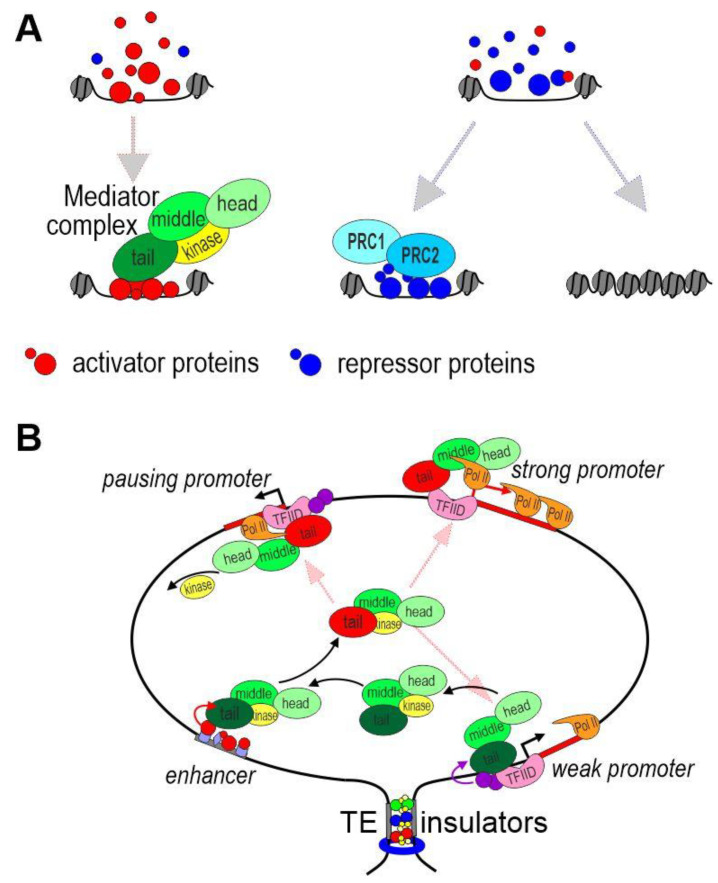
Model of promoter activation by an enhancer. (**A**) Activation or suppression of enhancers. The concentration of activators and repressors determines the fate of the enhancer in a particular nucleus. The mediator complex is recruited to the active enhancer. TFs can still bind to a repressed enhancer. In this case, Polycomb proteins play an important role in the suppression of enhancer activity. Alternatively, compaction of chromatin leads to dissociation of TFs from the enhancer. (**B**) Possible mechanism of functional interaction between an enhancer and promoters at a distance. Tethering elements or insulators form a chromatin loop that brings promoters into the active zone of the enhancer. The mediator complexes bind to the promoters located in the area of the enhancer. The level of transcription depends on the properties of a particular promoter.

**Figure 3 ijms-24-02855-f003:**
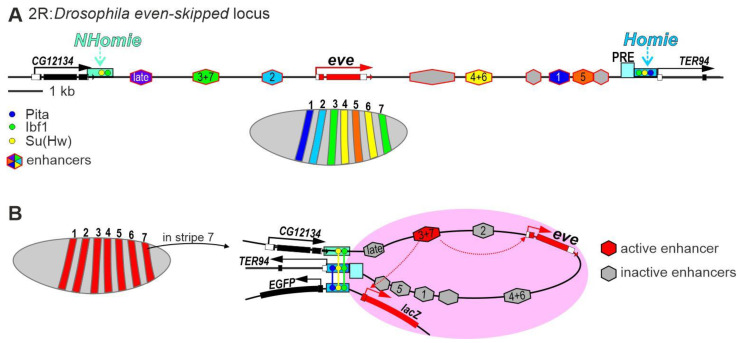
Model of transcriptional regulation of the pair-rule gene *eve* in early Drosophila embryos. (**A**) Schematic representation of the *eve* regulatory region that is flanked by the Homie and NHomie insulators. (**B**) Transcriptional activation model of the endogenous *eve* gene and the reporter transgene in the stripe 7 of early embryos. The interaction between the Homie and NHomie insulators forms a zone in which the activated *eve* enhancer can stimulate transcription of the endogenous *eve* promoter and the reporter gene promoter. Identical copies of the Homie insulator located in the endogenous *eve* locus and the transgene interact in head-to-head orientation, which brings only the reporter located on the head side of the insulator into the active *eve* enhancer zone.

**Figure 4 ijms-24-02855-f004:**
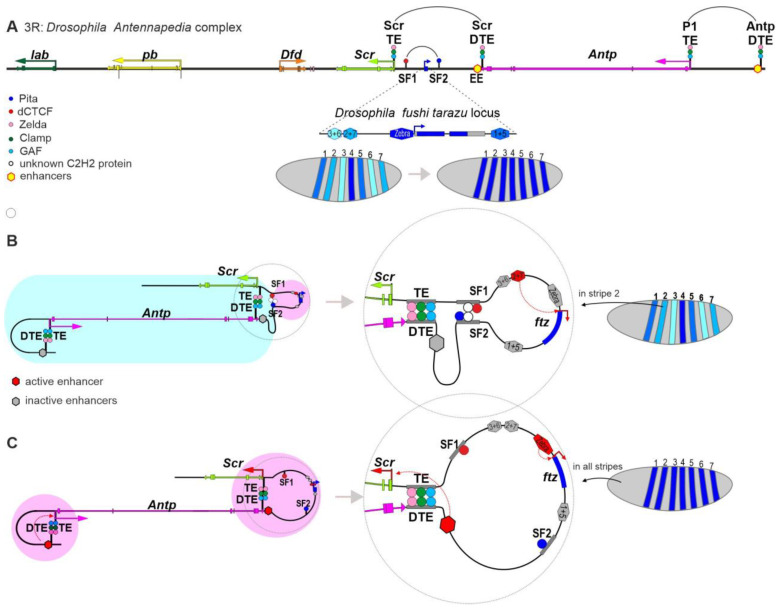
Model of transcription regulation for the pair-rule gene *ftz* and the *Scr* gene from ANT-C. (**A**) Schematic representation of ANT-C and the regulatory region of the *ftz* gene. Only the enhancers, insulators, and tethering elements described in the text are indicated. (**B**) Transcription regulation model of the endogenous *ftz* in stripe 7 of early embryos. Active and inactive chromatin zones are marked in pink and blue, respectively. The interaction between the SF1 and SF2 insulators allows for autonomic regulation of the *ftz* gene. (**C**) Transcription regulation model of the endogenous *ftz and Scr* genes in 5–6 h embryos. The interaction between the tethering elements allows for effective activation of the *Scr* and *Antp* promoters by their early enhancers.

**Figure 5 ijms-24-02855-f005:**
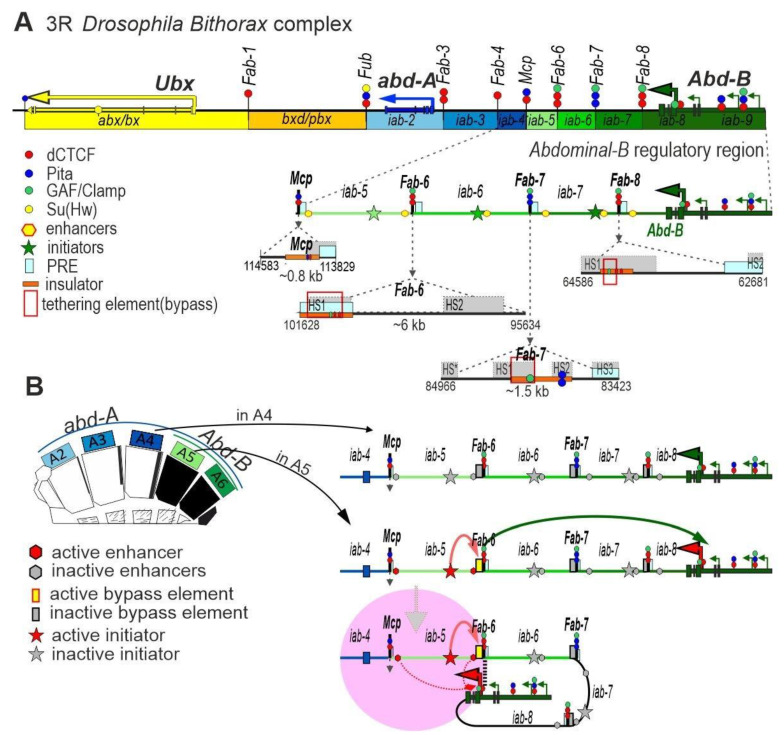
Boundaries organize enhancer–promoter interactions in the *Abd-B* gene of the BX-C. (**A**) Map of the BX-C showing the location of the three homeotic genes and the parasegment-specific regulatory domains. There are nine *cis*-regulatory domains (shown as colored boxes) that are responsible for the regulation of the BX-C genes and the specification of parasegments 5 to 13, which correspond to T3-A8 segments. The *abx/bx* (yellow) and *bxd/pbx* (orange) domains activate *Ubx*, *iab-2–iab-4* (shades of blue) activates *abd-A*, and *iab-5*–*9* (shades of green) activates *Abd-B*. Lines with colored circles mark chromatin boundaries. The dCTCF, Pita, and Su(Hw) binding sites at the boundaries are shown as red, blue, and yellow circles, respectively. (**B**) Model of *Abd-B* activation in A5/PS11. The active chromatin zone is marked in pink.

## Data Availability

Not applicable.

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
