# Peer review of "Mechanisms of Interaction between Enhancers and Promoters in Three Drosophila Model Systems"

_ijms, 2023, doi:10.3390/ijms24032855_

Round 1

Reviewer 1 Report

The authors have summarized the mechanisms of long-range interactions in the three well-studied key loci involved in the regulation of Drosophila development. The paper was well-written, and the topic is important.

I have two suggestions:

1. the authors should use an independent part to explain, why the fruit fly is a good model organism for studying the interaction between enhancers and promoters.

2. the authors should compare the mechanisms of long-range interactions in the three loci in Drosophila with the corresponding situation in mammals. 

To show, whether the studies on Drosophila can be well translated to medical usage. This will bring much broader interest.

Additionally, Drosophila should be written in italics.

Author Response

“the authors should use an independent part to explain, why the fruit fly is a good model organism for studying the interaction between enhancers and promoters”

We have discussed why the fruit fly is a good model in the Introduction and Conclusion sections.

“the authors should compare the mechanisms of long-range interactions in the three loci in Drosophila with the corresponding situation in mammals. To show, whether the studies on Drosophila can be well translated to medical usage. This will bring much broader interest.”

We have tried to emphasize in several places of the review that the mechanisms of long-range interactions in Drosophila and mammals are similar. How the results of the Drosophila study can be used in medicine are briefly discussed in the Conclusion section.

“Additionally, Drosophila should be written in italics.”

Fixed.

Reviewer 2 Report

Overall, this review article by Kyrchanova et al. is interesting. The authors have summarized research studies related to chromosome architecture in the nucleus and the formation of long-distance interactions between enhancers and promoters. The second part of the review describes three well-studied Drosophila regulatory systems at the eve locus, Bithorax, and Antennapedia gene complexes.   However, before publishing this manuscript, the authors must clarify and improve the following issues:   1) Missing citations: Line 50-52, Line 52-53, Line 57-58, Line 77-79, Line 208-210, Line 231-234, Line 237-238, Line240-241   2) Missing Figure 3 (repeated Figure 4): Without the figure, it was difficult to understand section 4 (Line 220-324)   3) In section 3 authors have described How activator proteins and mediator complexes interact with transcription initiation complex to explain enhancer-promoter communication but the authors have failed to explain it in the context of eve locus, Bithorax, and Antennapedia gene complexes regulatory systems (Section 4,5, and 6)

Author Response

“However, before publishing this manuscript, the authors must clarify and improve the following issues:   1) Missing citations: Line 50-52, Line 52-53, Line 57-58, Line 77-79, Line 208-210, Line 231-234, Line 237-238, Line240-241”

We inserted the missing citations

 “Missing Figure 3 (repeated Figure 4): Without the figure, it was difficult to understand section 4 (Line 220-324)”

We apologize for the problem with Figure 3, which was accidentally replaced with a copy of Figure 4.

“In section 3 authors have described How activator proteins and mediator complexes interact with transcription initiation complex to explain enhancer-promoter communication but the authors have failed to explain it in the context of eve locus, Bithorax, and Antennapedia gene complexes regulatory systems (Section 4,5, and 6)”

We explain potential mechanisms of enhancer-promoter regulation in the Drosophila model systems.

We also supplemented the corresponding Figures with model diagrams.

Round 2

Reviewer 2 Report

All my comments and suggestions have been addressed.